# Hearing-loss related variations in turn-taking time affect how conversations are perceived

Eline Borch Petersen *

ORCA labs, Lynge, Denmark

* eline.petersen@wsa.com

## Abstract

In conversations, interlocutors with hearing impairment (HI), initiate their turns with longer average delay and more variability than their normal-hearing (NH) conversation partners. This paper investigates whether third-party listeners were able to perceive this difference in turn-taking timing (denoted Floor-Transfer Offsets, FTOs) between NH and HI interlocutors. To this avail, the FTOs of segments of conversations were manipulated using four different schemes, such that the two interlocutors timed their turns as if 1) both were NH interlocutors conversing in quiet (NHQ) or 2) both were HI interlocutors conversing in noise (HIN). Two additional conditions were implemented to test if non-varying and faster FTOs affect the perception, manipulating conversations to have constant FTOs corresponding to 3) the median FTO of the NHQ condition (conNHQ) and 4) an even faster FTO of 50 ms (conLow). Forty-four participants rated conversations from the HIN condition as less natural, with poorer flow, and being more difficult to follow compared to conversations from the NHQ condition. Removing the FTO variability resulted in better ratings of flow (conNHQ vs NHQ), while further decreasing the average FTOs (conLow vs conNHQ) had no effect on the ratings. A detailed analysis revealed poorer ratings of flow for individual conversations with higher FTO median, but similar variability, while FTO variability alone did not affect ratings of flow. Together, the results indicate that listeners, and perhaps also conversation participants, are sensitive to small variations in turn-taking timing and that hearing loss may therefore affect the quality of conversations.

## Introduction

Conversations are an important part of our lives, but can be difficult to manage for people suffering from hearing loss. In a series of recent studies, it has been established that hearing impairment affects the dynamics of a conversation: In two-person conversations between a normal-hearing (NH) and a hearing-impaired (HI) interlocutor, it has been consistently found that the HI interlocutors are slower and more variable at initiating their turns [1–3]. However, the question arises whether the relatively

**Data availability statement:** The anonymized subjective ratings and a Matlab-script on the procedure used for implementing the FTO manipulation is available available via https://doi.org/10.5281/zenodo.15307719.

**Funding:** The author(s) received no specific funding for this work.

**Competing interests:** The author is an employee at the hearing-aid manufacturing company WS Audiology. This does not alter our adherence to PLOS ONE policies on sharing data and materials.

small differences between NH and HI interlocutors in the timing of their turns have any implications for how conversations are perceived. This study aims to investigate whether manipulating turn-taking timing affects how third-party listeners perceive a conversation.

When contributing to a conversation, the verbal utterance is often delivered in response to what your conversation partner has just said. However, when you start your turn is not completely without importance. The timing of a turn is denoted a Floor-Transfers Offsets (FTOs), i.e., the time between the conversation partner stop talking and you start, in other words, the time it takes for the floor to transfer from one talker to another. FTOs are measured in milliseconds and will be negative if you and your partner are talking in an overlap and positive if there is a pause between the conversation partner stops and you start. An example of FTO times from a short segment of a conversation is provided in Fig 1. Because FTOs vary between turns, they are often characterized by the median (FTO median) and variability (FTO inter-quartile range, IQR) of the turn-taking times accumulated across a conversation. In two-person conversations, interlocutors on average initiate their turns around 200 ms after their conversation partner stops speaking, a timing that is relatively stable across cultures and languages [4,5]. As it takes at least 450 ms to prepare and utter a single word [6], it is believed that accurate prediction of the turn-ending cues provided by the conversation partner is important to deliver a timely response within 200 ms [7–9]. Hence, when observing that, relative to NH interlocutors, HI talkers conversing in quiet are on average 80 ms slower (FTO median) and 180 ms more variable (FTO IQR) at initiating their turns [3], it could be speculated that this is caused by the HI interlocutors being less able to utilize the turn-ending cues provided by their conversation partner. Indeed, it was observed that when restoring audibility for the HI interlocutors, through hearing aids, their median FTO was reduced to the same level as the NH interlocutors, while the difference in FTO IQR was halved, to 90 ms [3].

In the presence of background noise, the FTO median and IQR increase for both NH and HI interlocutors; more so when interlocutors are seated in different rooms [10] than when seated face-to-face [2]. However, the presence of noise affected the turn-taking timing of the HI interlocutors additionally, as their FTO IQR increased by further 99 ms on top of the increase already seen for NH when conversing in noise [2].

Having established that impaired hearing affects the turn-taking timing, it is relevant to consider how this could affect the conversation partner. In face-to-face conversations, (NH) interlocutors' felt reduced social connectedness with conversation partners who were slower at initiating their turns [11]. This observation was also confirmed with third-party listeners: When doubling the individual FTOs, listeners rated the conversation partners as less socially connected than for conversations with the original FTOs and with FTOs reduced to one-fifth of the original length [11]. In [11] the FTOs were manipulated by scaling them up and down, however most studies manipulate FTOs by simply adding a delay, finding that this negatively affects how people perceive conversations [12,13]. Although delaying responses (larger FTO median) is relevant for evaluating telecommunication, our previous studies observe

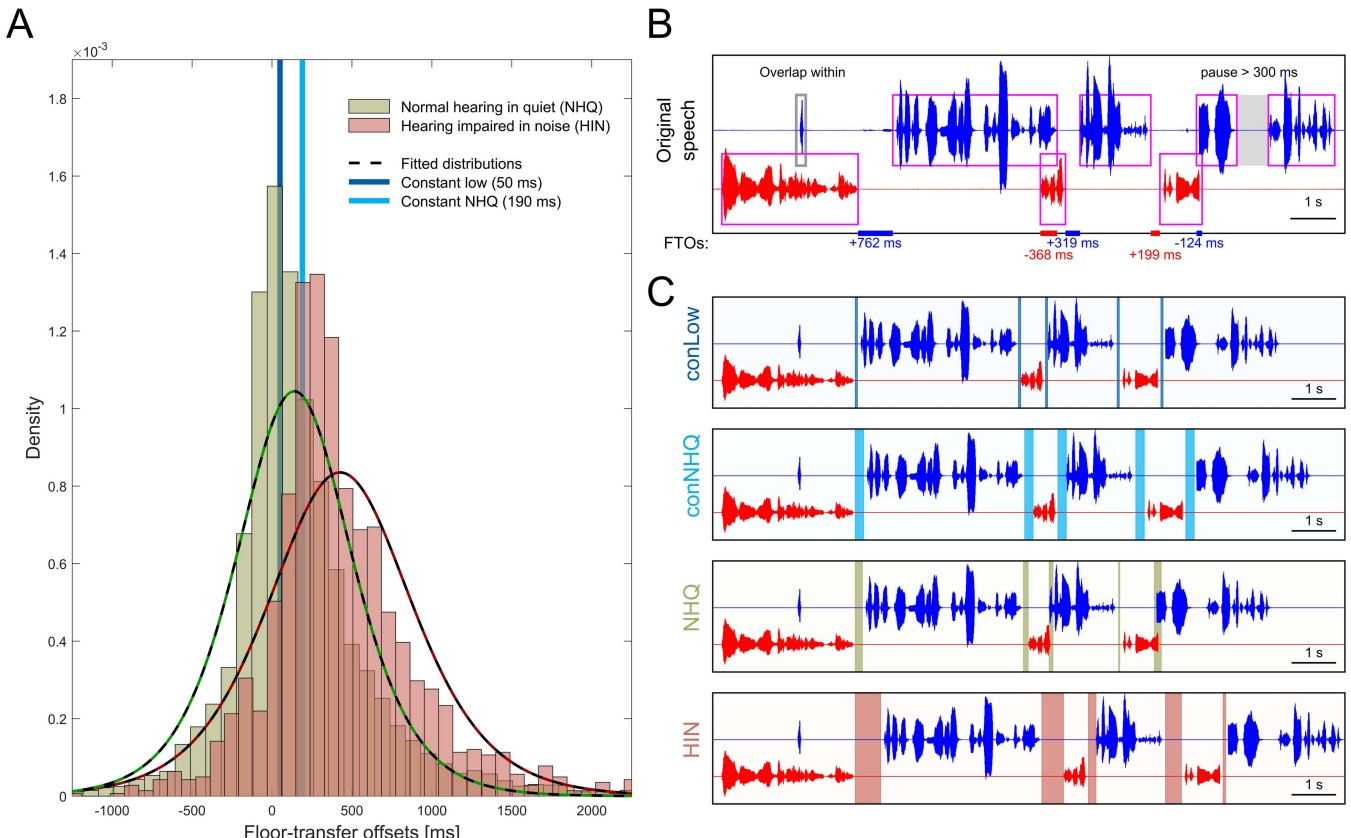

**Fig 1. Floor-transfer offset (FTO) distributions and examples of conversation manipulations.** A) Floor-transfer offsets (FTO) for normal-hearing interlocutors conversing in quiet (NHQ, green bars) and hearing-impaired interlocutors conversing in noise (HIN, red bars), overlaid with the best fitted logistic distribution (dotted black and colored lines) from which FTO values were drawn when manipulating the FTO values of a conversation segment. Vertical lines indicate the constant FTO values of 190 ms (conNHQ, light blue) and 50 ms (conLow, dark blue) used for the manipulation schemes with constant FTO values. B) An example of an original conversation between two normal-hearing talkers (red and blue) and the corresponding detected voice activity (magenta boxes). When manipulating the FTO values, the timing of "overlaps within" (grey box) was kept and the duration of pauses longer than 300 ms was halved (shaded grey area). The lines on the x-axis indicate FTOs between the two talkers with the color indicating which talker the FTO value (provided below the line) belongs to (blue or red). Positive FTO values occur when there is a pause between the two talkers, negative values when a new turn is started in overlap with the turn of the current speaker. C) Examples of the four FTO manipulation conditions. The new FTOs are highlighted with shaded areas, using the same color coding as in A). For conLow and conNHQ, all FTO values were the same throughout the conversation. For NHQ, the FTO times were drawn at random from the logistic function describing the NHQ FTOs. The FTO values used in the HIN condition were transformed based on values drawn from the NHQ.

that in conversations, not only FTO median, but also the FTO variability increase with increased communication difficulties (impaired hearing and/or background noise). Hence, the potential effect of FTO variability on the perception of conversations should not be neglected. The purpose of the current study is to investigate who naturally occurring variations in FTO median, as well as in the often overlooked FTO IQR, affects how conversations are perceived.

The current study will investigate 1) whether the relatively small differences between NH and HI in the timing of turns can be perceived by third-party listeners, and 2) whether faster turn-taking is perceived as being 'better'. To this avail, we manipulated the timing of turns in a corpus of two-person conversations [14]. The study included four different conditions where FTO was manipulated. In the first two conditions, FTOs were randomly chosen from the FTO distributions of NH interlocutors conversing in quiet (NHQ condition) and HI interlocutors conversing in noise (HIN condition); in these

conditions, FTO was variable as it would be in a "normal" conversation. The second two conditions tested if non-varying and faster FTOs affected perception and evaluation by manipulating conversations to have two different constant FTO values: One where all turns were initiated with the median FTO value 190 ms, corresponding to that of NH interlocutors conversing in quiet (conNHQ condition) and one with an even faster constant FTO of 50 ms (conLow condition).

Participants were asked to listen to short conversations manipulated with one of the four FTO manipulation schemes described above and afterwards subjectively rate three aspects of the conversation: 1) How natural they found the conversation (denoted naturalness), 2) how free-flowing they found the conversation (denoted flow), and, 3) how easy they found it to follow the conversation and the information being exchanged (denoted ease-of-following). We hypothesized that conversations with faster and less variable turn-taking times would be perceived as 'better' (conLow > conNHQ > NHQ > HIN), meaning higher ratings of naturalness, conversational flow, and conversations being easier to follow. As the FTO manipulation directly affects the flow of the manipulated conversations, we also hypothesized that the flow ratings would be more sensitive to the FTO manipulation than the ratings of naturalness and ease-of-following.

## Materials and methods

### Choosing relevant FTO values

To evaluate to effect of FTO median and IQR on the subjective evaluations of conversational flow, naturalness and ease-of-following, the turn-taking timing of shorter segments of conversations were manipulated according to four different manipulation schemes: Two with constant (conNHQ, conLow) and two with varying FTOs drawn from distributions representing the FTO times of NH conversing in quiet (NHQ) and the FTO times of HI conversing in noise (HIN).

From an online corpus of conversations between 19 pairs of NH interlocutors solving a spot-the-difference task in their native language of Danish in quiet surroundings [14], voice activity detection was performed, and the FTO values representing the NHQ condition was extracted (see Manipulating FTOs of conversations for more detail). An FTO distribution was generated by fitting a logistic distribution to the histogram of the extracted FTO times (NHQ in Fig 1A), resulting in a distribution with the parameters $\mu = 139.8$ ms and $\sigma = 239.3$ ms.

A similar procedure was followed with the FTO times extracted from 11 HI interlocutors wearing hearing aids while solving the same spot-the-difference task in collaboration with a NH interlocutor in the presence babble noise presented at 70 dB SPL [2]. The logistic distribution best describing the HIN FTO values had the parameters $\mu = 454.7$ ms and $\sigma = 382.5$ ms (HIN in Fig 1A).

For the manipulation scheme conNHQ, the turn-taking of the conversation were always set to the median value of FTO distribution of the NHQ condition, which was 190 ms (Fig 1A). For the conLow condition all turns were manipulated to start 50 ms after the conversation partner stopped (Fig 1A).

### Manipulating FTOs of conversations

The four FTO manipulation conditions described above were applied to a subset of the conversations from the corpus from which the NHQ FTO values were extracted [14]. As the interlocutors were positioned in different rooms, the recording does not contain crosstalk, making them ideal for manipulation. From the 19 pairs in the corpus, four pairs were selected because i) of the high number of turns per conversation, ii) they had subjectively distinguishable voices, and iii) they represented different gender combinations (pair 11 and 20: male-female, pair 16: male-male, pair 19: female-female). Three conversations were available for each of the four pairs in which they solved one spot-the-difference task in quiet surroundings and in their native language of Danish. For each pair, a total of eight time-intervals were selected across the three conversations in which there were a high number of turn changes. The length of these segments varied slightly to avoid ending the segment mid-sentence. On average, the segments were 32.2 seconds long (range: 29.0–36.9 seconds) and contained an average of 17.9 turn changes (range: 12–25 turns).

For each of the conversation segments, voice activity detection was performed using the Kaldi method [15] with 5-ms windows being marked as speech when more than 15% of the log-energy exceeded the median log-energy plus 5 dB [3]. Afterwards, the voice activities of both conversation partners were fed into a communicative state classifier [16] to determine the start and end of each utterance and classify them as turns or as "*overlaps within*" (see Fig 1B for example). Turns are defined as sequences of connected inter-pausal units (speech) from one interlocutor, including pauses, surrounded by floor transfers. It is the durations of these floor transfers that were manipulated in the current study. An "overlap within" is speech uttered completely within another interlocutor's turn and do not result in a floor transfer. As will be described further in the following, the relative position of the overlaps within were kept, ensuring that the conversions were as natural as possible.

Fig 1B shows an example of an original conversation with the corresponding detected voice activity. Based on the classification of voice activity into turns and overlaps within, each conversation segment was manipulated in the following way: A silent period of 200 ms was added to the beginning of each conversation. For each turn: 1) The loudness was equalized by scaling the level of the inter-pausal units (turns excluding pauses) to a fixed root-mean-square value. 2) If the turn contained pauses longer than 300 ms, the duration of the pause was halved until its duration was shorter than 300 ms (Fig 1B, grey area). 3) If overlaps within were present, the loudness of these were equalized, as in 1), and the overlap within added to the conversation with the same timing as in the original conversation (Fig 1B, grey box). 4) After the end of the turn, a new FTO value was chosen according to one of the four FTO manipulation schemes. The start of the next turn was set accordingly, either as a constant value (conNHQ and conLow) or drawn randomly from the distribution describing the NHQ and HIN conditions. For the latter two, the next turn can happen either in a gap/pause (positive FTO) or in an overlap (negative FTO) with the preceding turn. The above steps 1–4 were repeated for each turn within the conversation segments.

Finally, the manipulated conversation segment was saved as a single-channel wav-file sampled at 48 kHz. Note that although the conversation segments originate from interactions between two NH interlocutors conversing in quiet, the conversations in the NHQ condition were also manipulated in the manner described above. An Matlab-script illustrating the implementation of the manipulation steps outlined above can be found on Zenodo (https://doi.org/10.5281/zenodo.15307719).

As evident from Fig 1A, the distribution of FTO values for the NHQ condition is very similar to the distribution for the HIN condition, with a so-called overlapping index between the two distributions of 69% [17]. This means that if FTO values are randomly choose from the two distributions, they will often be very similar, and the manipulation would not successfully mimic the observed difference between NHQ and HIN. To mitigate this, conversation segments in the NHQ and HIN conditions were manipulated in pairs in the following manner: For one conversation segment in the NHQ condition, FTO values were randomly chosen from the NHQ distribution and applied. Afterwards, the NHQ FTO values were transformed by scaling and shifting them such that they fit the HIN distribution, and applied to another conversation segment. The third-party listeners were presented with both conversations during the experiment, in a randomized order. In this manner, it is ensured that the FTO values used in the NHQ and HIN conditions were different, while still drawing them randomly from the distribution.

## Experimental setup

In the experiment, each participant was presented with the 32 conversation segments (eight segments from four conversation pairs), i.e., each conversation segment was only presented once. The four FTO manipulation conditions (conLow, conNHQ, NHQ, and HIN) were applied to eight conversations each, such that each condition were applied to two randomly chosen conversation segments from each of the four pairs. To avoid basing this study on the evaluation of just eight instances of randomly chosen NHQ FTO values, and the corresponding scaled FTO values used for the HIN condition, ten lists were generated in which different random NHQ FTO values, and consequently HIN FTOs, were applied. For the eight instances of the NHQ condition within a list, new FTO values were selected at random. Hence, a total of 80 different

renderings of the NHQ-manipulated conversation were generated across the ten lists, drawing from the same FTO distribution, but resulting in different FTO values being applied. Which of the ten lists a participant was presented with, was selected based on the second of the universal time at which the experiment was initiated and can as such be reasonably considered as randomly chosen.

The experiment was implemented in PsychoPy [18] and exported to run online using the Pavlovia interface (Open Science Tools, Nottingham, UK). Upon initiating the experiment, the participant was asked to provide their age, their level of Danish (native, second language, or not Danish speaking), their gender (male, female, or other), and their hearing status (normal hearing, hearing impaired not using hearing aids, or hearing-impaired using hearing aids). To start the experiment, the participants had to accept a written informed consent declaration, otherwise the experiment was terminated. After informed consent was given, a short conversation segment (not used in the experiment) was presented with the instruction to adjust the volume of their computer to a comfortable level. Participants were instructed to listen through headphones, but no check was conducted to ensure that participants complied with this request. Next, participants were presented with the written instruction of the experiment, including explanations of the three attributes of the conversation they were asked to rate after each conversation ended.

Thereafter, the 32 FTO-manipulated conversations were presented in a random order. After having listened to each conversation, participants were asked to provide three subjective ratings, one after the other, on a continuous visual analogue scale with five anchor points. For each attribute they rated, the explanation given in the written instruction was displayed on the screen. First, participants were asked to rate '*How natural did you find the conversation?*' with the outer anchors specifying '*Completely natural*' to '*Completely unnatural*' and the explanation '*Natural means whether the conversation sounds like a spontaneous exchange of information between the two talkers*'. Secondly, participants were asked to rate '*How free-flowing did you find the conversation?*' with the outer anchors specifying '*Not at all free-flowing*' to '*Completely free-flowing*' with the explanation '*The question refers to whether the talkers had a good flow in their conversation without long pauses and/or long overlaps in their speech*'. Finally, participants were asked to rate '*How easy was it for you to follow the conversation?*' with the outer anchors specifying '*Very difficult*' to '*Very easy*' with the explanation '*The question refers to whether you are able to follow what is being said and understood the information exchanged without concentrating too much*'. For all three rating-scales, the mid-point was labelled '*neither/nor*'. From here on, these three ratings will be referred to as ratings of 'naturalness', 'flow' and 'ease-of-following', respectively.

After every eighth conversations, the participant was able to take a break of self-administered length. The entire experiment took around 35 minutes to complete.

## Participants

Participants were recruited among employees at WS Audiology within the period 09/22/2024–11/26/2024. A total of 49 people completed the experiment, however only data from the 44 participants indicating they had normal hearing were analyzed. The 44 participants had a mean age of 43.6 year (std 14.2, range 19−73) and consisted of 20 females (45.5%) and 24 males (54.5%). Male and female participants did not differ in age (t(42) = −0.85, p = 0.4)

The Research Ethics Committee of the Capital Region of Denmark provided an ethical waiver (waiver number H-20068621) for the current study, as they did not find it necessary for a full ethical approval.

## Statistical analysis

The three subjective ratings of naturalness, flow, and ease-of-following were averaged across the eight repetitions for each participant and the effect of FTO condition was analyzed using mixed-effects regression models with the lme4 package for R [19]. Besides the fixed effects of FTO condition (conLow, conNHQ, NHQ, and HIN), the models also included the fixed effect of gender (female and male) and the interaction effect between FTO condition and gender. Subject was added as a random effect. Finally, post-hoc analysis of significant factors was conducted using the function ls_means from

the lmerTest package [20]. For all statistically significant effects of the post-hoc testing, the 95% confidence intervals (CIs) of the regression coefficients are provided.

It was further investigated how sensitive the subjective rating of flow was to the FTO values, with the FTO median and IQR value extracted from individual NHQ and HIN conversations. Using Spearmans' correlation analysis across all conversations and participants, the relationship between the subjective rating of flow and FTO median and IQR was investigated. Wilcoxon rank-sum tests were used to investigate whether subsets of conversations, extracted to vary in only one of the FTO parameters, differed in the ratings of flow.

## Results

### Effect of FTO condition on subjective ratings

The average ratings of naturalness, flow, and ease-of-following are shown in Fig 2A for each participant and condition, with horizontal lines indicating the statistically significant (post-hoc) differences between FTO manipulation conditions. An overview of the statistical analyses is show in Table 1.

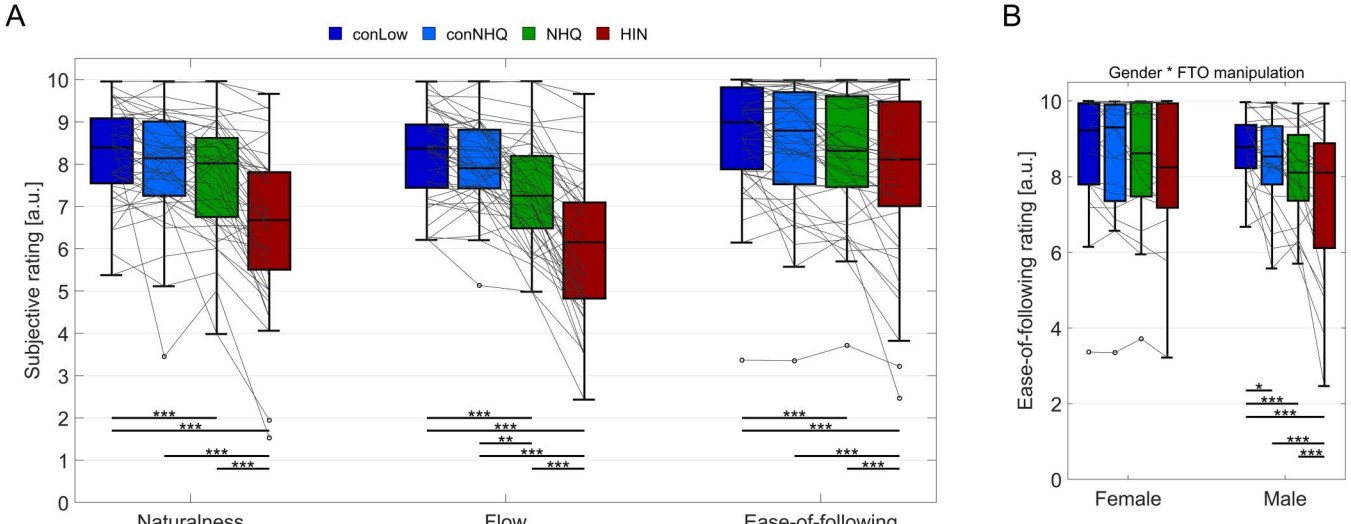

**Fig 2. A) Subjective ratings of naturalness, flow and ease-of-following for the four FTO manipulation conditions.** Boxes indicate the 25th to 75th percentile, with horizontal lines indicating the median. Whiskers extend to data points larger than 1.5 times the interquartile range from the bottom or top of the box, with outliers indicated in black open circles. Average ratings for single participants are shown in grey lines. Horizontal lines and asterisks indicate statistically significant differences between conditions resulting from post-hoc tests (* $p < 0.05$, ** $p < 0.01$, *** $p < 0.001$). B) Highlight of the statistically significant interaction between gender of the participant and the rating of ease-of-following.

**Table 1. Statistical test values and corresponding p-values for the three subjective rating categories for the two fixed effects (FTO condition and gender) and their interaction.** Statistically significant effects are highlighted in bold.

| Rating Fixed effect | Naturalness | | Flow | | Ease-of-following | |
|---|---|---|---|---|---|---|
| | **F-stats** | **p-value** | **F-stats** | **p-value** | **F-stats** | **p-value** |
| FTO Condition $F_{(3,114)}$ | **25.3** | **< 0.001** | **44.9** | **<0.001** | **13.8** | **<0.001** |
| Gender $F_{(1,38)}$ | 1.0 | 0.3 | 0.9 | 0.4 | 0.01 | 0.9 |
| FTO condition * Gender $F_{(3,114)}$ | 1.2 | 0.3 | 1.4 | 0.2 | **5.1** | **<0.01** |

 

Subjective ratings of the **conversation naturalness** were significantly affected by FTO condition (p < 0.001, Table 1). The post-hoc analyses, see Fig 3A, revealed that the ratings of naturalness for the conLow condition were higher than for NHQ (t(117) = 2.6, CI: 0.13–0.9, p < 0.001) and HIN (t(117) = 8.4, CI: 1.3–2.1, p < 0.001), but did not differ significantly from the ratings of conNHQ (p = 0.15). In comparison to the HIN conditions, the ratings of naturalness were significantly higher for conversations manipulated with the conNHQ (t(117) = 6.9, CI: 0.98–1.7, p < 0.001) and the NHQ conditions (t(117) = 5.7, CI: 0.74–1.6, p < 0.001).

The **rating of flow** was also affected by FTO condition (p < 0.001, Table 1). Conversations were perceived to have better flow when manipulated with conLow compared to NHQ (t(117) = 4.1, CI: 0.4–1.2, p < 0.001) and HIN (t(117) = 10.7, CI: 1.7–2.5, p < 0.001), but did not differ significantly from the ratings of conNHQ (p = 0.3). The conversations with the NHQ FTOs were rated as having better flow than the conversations from the HIN condition (t(117) = 9.7, CI: 1.6–2.2, p < 0.001), but poorer flow than the conversations in the conNHQ condition (t(117) = 3.0, CI: 0.21–0.99, p < 0.01).

Finally, the **ease-of-following** was affected by the FTO manipulation scheme (p < 0.001, Table 1) and by an interaction between FTO manipulation and gender (p < 0.01, Table 1). The visualization of the significant interaction between gender and FTO condition (p < 0.01, Table 1) in Fig 2B indicate that ratings from the female participants exhibit a ceiling effect across FTO conditions. The post-hoc tests showed that the ease-of-following ratings made by the females were not affected by FTO manipulation (all p's > 0.1). However, the male participants rated it easier to follow the conversations manipulated by conLow relative to conNHQ (t(114) = 2.1, CI: 0.01–0.75, p = 0.03), and easier for the NHQ condition than

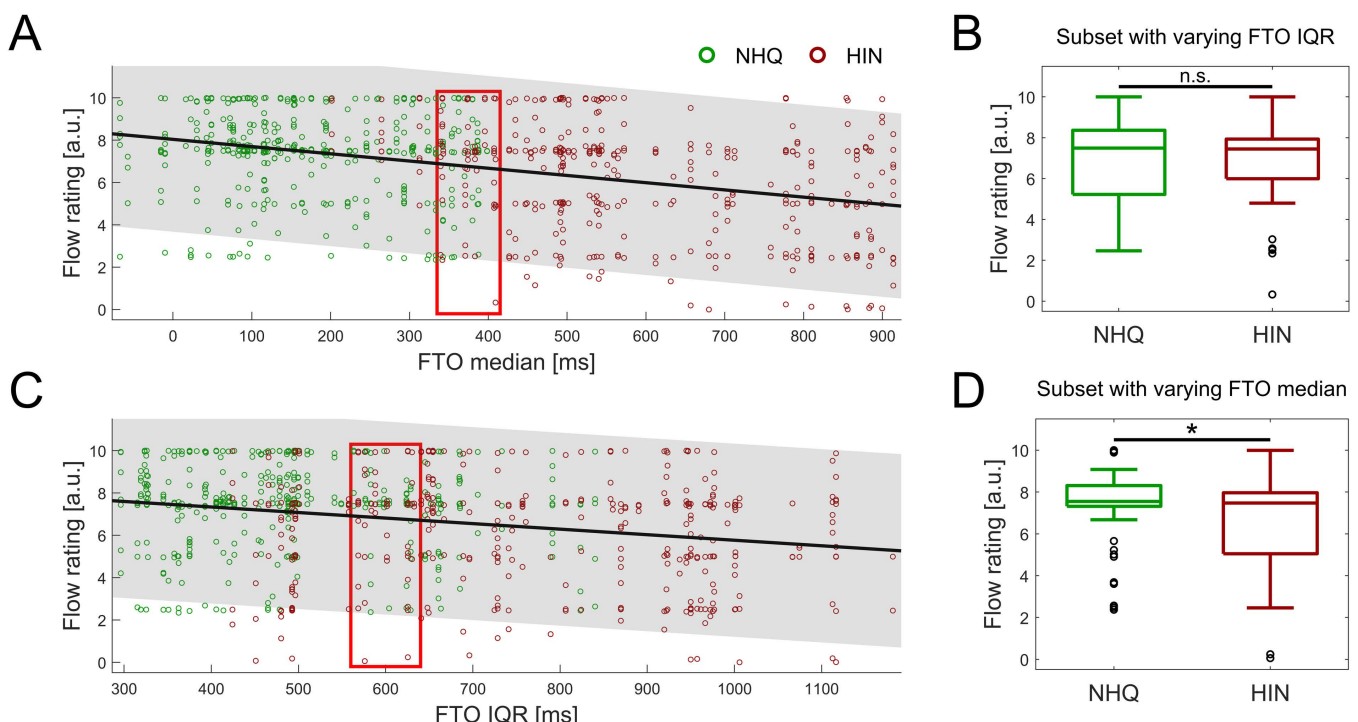

**Fig 3. Relation between FTO values and rating of flow.** A+C) Correlation between FTO median and IQR of each conversation manipulated with NHQ (green circles) and HIN (red circles) and the corresponding subjective rating of conversational flow. Each circle indicates the rating of one conversation made by one participant. The black line indicates the best-fitted linear relation between FTO value of the conversation and the corresponding flow rating, with the shaded grey area highlighting the 95% confidence interval of the regression line. The red boxes indicates the interval from which a subset of conversations with constant FTO median (A) or IQR (C), were extracted. B+D) Subjective ratings of flow from the subset of NHQ and HIN conversations with (B) similar FTO median and differing FTO IQR and with (D) similar FTO IQR and differing FTO median. Vertical lines indicate the result of the comparative Wilcoxon rank-sum tests between the NHQ and HIN conditions (* p < 0.05, n.s. non-significant).

the HIN (t(114) = 3.6, CI: 0.3–1.02, p < 0.001). The difference between conNHQ and NHQ only approached significance (t(114) = 1.9, CI: −0.01–0.71, p = 0.057).

## Ratings of flow for individual conversations

As hypothesized, and as evident from the results in Table 1 and Fig 2A, the ratings of conversational flow were over-all more sensitive to the FTO manipulation scheme than ratings of naturalness and ease-of-following. Hence, a more in-depth analysis was made of the relation between the flow ratings of individual conversations and the FTO parameters (median and IQR) applied to the specific conversation segment. This analysis focused on the conversations from the NHQ and HIN conditions, because they vary in both FTO parameters. As seen in Fig 3, conversations with larger FTO medians were perceived as less flowing (r = −0.37, p < 0.001, Fig 3A) and similarly conversations with more variability (larger FTO IQR) also was lower perceived flow (r = −0.23, p < 0.001, Fig 3C). However, as the conversations from the HIN manipulation scheme has both larger FTO median values and larger FTO IQR values, the separate effects of manipulating the FTO median and IQR cannot be deduced from the two significant correlations shown in Fig 3A, C.

To investigate the effect of FTO IQR, independently of FTO median, a subset of 214 conversations with broadly similar median FTOs, ranging from 335−415 ms (red box in Fig 3A), were extracted. This median FTO interval was selected by extensively searching for the interval containing the highest number of conversations in which the FTO median did not differ significantly between the NHQ and HIN conditions, with the constraint that the number of NHQ and HIN conversation should not differ by more than 20. Of the 214 selected conversations, 98 were manipulated with the NHQ scheme and 116 with the HIN scheme. A rank-sum test confirmed that the FTO median did not differ between NHQ and HIN within this subset (Wz = −1.6, p = 0.1), while the FTO IQR of the HIN conversations (mean of 413 ms) was higher than the NHQ FTO IQR (mean of 665 ms, Wz = −9.7, p < 0.001). Despite differing in FTO IQR, the ratings of flow for this subset of conversations did not differ between the NHQ and HIN conversations (Wz = 0.3, p = 0.75, Fig 3B). This suggests that differences in the FTO IQR do not affect the perception of conversational flow.

In a similar manner, 202 conversations with similar FTO IQR values, range 560−640 ms, see red box in Fig 2C, were extracted to investigate the effect of FTO median on the ratings of flow. Of these conversations, 94 were manipulated with NHQ and 108 of them with the HIN scheme. Rank-sum tests confirmed that the NHQ and HIN conversation did not differ in FTO IQR (Wz = 0.2, p = 0.8), while the average 224 ms FTO median of NHQ conversations were significantly lower than the 570 ms FTO of the HIN (Wz = −10.6, p < 0.001). A significant 0.48 point increase in average rating of conversational flow was observed for the NHQ-manipulated conversations compared to those manipulated with the HIN scheme (Wz = 2.0, p = 0.045, Fig 3D), suggesting that the third-party listeners perceived higher median FTO values, manifesting as longer pauses between turns, as causing a poorer conversational flow.

## Discussion

In summary, the result showed that third-party listeners perceived conversations manipulated with FTO values based on HIN as being less natural, having poorer flow, and being more difficult to follow (Fig 2A) compared to NHQ. Removing the turn-taking variability (NHQ vs conNHQ) resulted in better ratings of conversational flow, whereas reducing the median FTO (conNHQ vs conLow) did not affect any ratings. In contrast to these results, relating the ratings of flow of individual conversations to the FTO median and IQR applied to the segment, showed that the perception of flow was only affected by the FTO median (Fig 3D), and not by FTO IQR (Fig 3B).

### FTO manipulation partly affects subjective ratings

It was hypothesized that third-party listeners would perceive conversations with larger FTO median and FTO variability (IQR) as less natural, having worse flow and being less easy to follow. The individual ratings partly confirmed this hypothesis (Fig 2A): Conversations with a low, non-varying FTO median (conLow) were rated higher in naturalness, flow and

ease-of-following compared to conversations with varying FTO values (NHQ and HIN). Likewise, the increased FTO median and IQR between NHQ and HIN also resulted in poorer ratings of all three parameters (Fig 2A). However, the hypothesis that faster FTOs would be perceived as 'better' was not confirmed, since the ratings did not differ between conversations from the conLow and conNHQ conditions. Hence, almost quadrupling the FTO median, from 50 ms to 190 ms, did not impact the ratings made by the third-party listeners. Adding variability to the conversational (conNHQ vs NHQ) did not affect the ratings of ease-of-following or naturalness, but did result in a poorer perceived conversational flow.

Based on these observations it is not possible to definitively ascribe the variations in perceived naturalness, flow, and ease-of-following specifically to FTO median or variability. However, the results clearly show that when FTO parameters match those of HI interlocutors conversing in a noisy environment (HIN), these conversations are perceived as less natural, having worse flow, and as being more difficult to follow. This suggests that the alterations in the FTO median and IQR observed for interlocutors with impaired hearing can affect how third-party listeners, and maybe even the conversation partner, perceive the quality of the conversation.

For the ratings of how easy it was to follow the conversations, an interaction effect was observed between the FTO manipulation schemes and the self-identified gender. When male listeners rated the ease-of-following the conversations, significant differences were observed between all pairwise comparisons of the FTO conditions (Fig 2B), while the ratings from the female listeners were not affected by FTO manipulation at all. We initially hypothesized that the subjective ratings of flow would be most sensitive to the difference in FTO manipulation, however it seems that male third-party listeners, surprisingly, show more sensitivity to the FTO conditions in their ratings of how easy it is to follow the conversation. This difference in ratings between male and female listeners cannot be readily explained based on existing literature on the effect of gender on speech perception.

## FTO median, not IQR, affects perception of conversational flow

Although significant effects of FTO manipulation schemes were observed on the ratings averaged across the eight repetitions per condition for each participant, it is evident from Fig 2B that the ratings vary substantially between individuals. This is especially true for the ratings provided for the NHQ and HIN manipulation schemes, most likely because the individual conversations in these two conditions vary in the FTO times. To investigate whether listeners were sensitive to the FTO times applied to a single conversation segment, the flow ratings for all listeners and conversation segments in the NHQ and HIN conditions were extracted and correlated with the FTO median and IQR applied to the specific conversation segment (Fig 3A, C). Although the ratings of flow decreased significantly for conversation segments with higher FTO median and higher FTO IQR (Fig 3A, C), the experimental design introduced a confound of FTO median and IQR. as both are increased in the HIN conversations relative to the NHQ. Hence, to investigate the effect of FTO median independently of FTO IQR, a subset of NHQ and HIN conversations were extracted in which one FTO parameter did not differ between conditions. From the extracted conversations, it is evident that the ratings of flow for conversations with similar FTO median and differing FTO IQR did not differ significantly (Fig 3B), while the flow-rating of the subset of conversations with different FTO median but similar FTO IQR did (Fig 3C). This suggests that FTO median affects flow ratings more than FTO IQR does.

Hence, the results of the current study present two contradicting observations: Firstly, while no effect of faster FTO median was observed between conLow and conNHQ (50 ms vs 190 ms, Fig 2A), significantly better rating of flow was observed for the subset of NHQ conversations with faster FTO median compared to a subset of HIN conversations with slower FTO median and similar FTO IQR (Fig 3D). Although a fast turn-taking times result in an objectively more efficient conversations, it is possible that the 140 ms increase in FTO median between conLow and conNHQ was not sufficient to affect the subjective ratings of flow, whereas the 346 ms difference in average FTO median between the subset of NHQ and HIN conversations was. It is also likely that humans are more sensitive to worse, rather than better, flow (relative to NHQ), as longer pauses can signal communication difficulty on the part of the (current) talker and requires the conversation partner to alter their way of communicating to ensure that the conversation can continue successfully.

The second discrepancy observed in this study is in the effects of FTO IQR, where the presence of variability reduced the ratings of flow between conNHQ and NHQ (Fig 2A), while no effect of FTO IQR was observed between the subset of NHQ and HIN conversations with similar FTO median (Fig 3C). One explanation for this difference could be that, despite aiming to apply similar FTO medians in the NHQ and conNHQ conditions, it is obvious from Fig 3A that the median FTO of NHQ vary between individual conversations. This is a consequence of manipulating shorter conversation segments, containing 12–25 turns each, by randomly drawing FTO value from the NHQ distribution. Hence, the observed difference between conNHQ and NHQ could be driven by the difference in FTO median between individual NHQ conversations and conNHQ, rather than a difference in FTO IQR. More care could of course have been taken to ensure a more reliable FTO median in the NHQ condition, however this could require choosing FTO values in a non-random fashion.

In a previous study, the effect of manipulating turn-taking times, by presenting the original conversations (FTO mean = 278 ms), as well as conversations where the individual turn-taking times were doubled (FTO mean = 557 ms) or shortened to one-fifth of the original (FTO mean = 56 ms) was investigated [11]. Third-party listeners perceived the conversation partners as enjoying the conversation less and being less connected with every step of increasing the FTO times [11]. The mean FTO times reported by Templeton correspond approximately to the median FTO times applied in the current study, further confirming that FTO times are related to how listeners perceive conversations [11]. However, it should be noticed that Templeton et al. did not consider that scaling the FTO times affects the variability, as well as the mean FTO times (assuming that the FTO times are not distributed equally around zero, evident from Fig 1A). When comparing between studies it should be noted that Templeton manipulated only six conversation segments, containing 10 turns extracted from spontaneous conversations, whereas the 32 segments consisting of between 12–25 turns used in the current study were extracted from conversations driven by solving a task. Despite the difference in the nature of the conversations, the results from the current study, compliments the findings by Templeton that, the FTO median times seem to affect the third-party perception of conversations more than the FTO variability.

Although the results of the current study suggest that longer FTO times result in less natural and flowing conversation, a previous study have found that longer FTOs are not unanimously perceived as 'worse' and that ratings also depend on the relationship of the interlocutors as well as the aim/nature of the conversation [21]. The conversations evaluated in the current study were excerpts from task-solving dialogues in which the interlocutors exchanged information to identify differences between images as fast as possible. The third-party listeners were informed about the nature of the conversations in the written instruction of the experiment, and they could therefore have been biased to rate faster exchanges as more natural and flowing. Had the conversations involved topics of a serious or personal nature, silence between turns might have been more positively perceived as they can convey internal reflection and active listening [21].

In conclusion, the current study showed that third-party listeners perceive conversations where turns are timed as NH interlocutors conversing in quiet as more natural, better flowing, and easier to follow than conversations where turns are timed as HI conversing in noise. Subsequent analyses of the individual conversations and ratings showed that this difference is mainly driven by differences in the median FTO, and not the variability (FTO IQR). Taken together, this could indicate that the slower turn-taking times observed for hearing-impaired interlocutors, both in the presence of noise, as well as when conversing without hearing aids in quiet, could affect, not just the speed of the conversation, but also how others perceive the conversation and potentially how socially connected conversation partners feel when engaging in conversations with HI interlocutors.

## Author contributions

**Conceptualization:** Eline Borch Petersen.

**Data curation:** Eline Borch Petersen.

**Formal analysis:** Eline Borch Petersen.

**Investigation:** Eline Borch Petersen.

**Methodology:** Eline Borch Petersen.

**Project administration:** Eline Borch Petersen.

**Visualization:** Eline Borch Petersen.

**Writing – original draft:** Eline Borch Petersen.

**Writing – review & editing:** Eline Borch Petersen.

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
