## [Decision Letter · Decision Letter 0]

22 Apr 2025

PONE-D-25-12104Hearing-Loss Related Variations in Turn-Taking Time affect how Conversations are PerceivedPLOS ONE

Dear Dr. Petersen,

Thank you for submitting your manuscript to PLOS ONE. After careful consideration, we feel that it has merit but does not fully meet PLOS ONE’s publication criteria as it currently stands. Therefore, we invite you to submit a revised version of the manuscript that addresses the points raised during the review process.

We look forward to receiving your revised manuscript.

Kind regards,

Niravkumar Joshi

Academic Editor

PLOS ONE

Journal Requirements:

The author is an employee at the hearing-aid manufacturing company WS Audiology

Reviewers' comments:

Reviewer's Responses to Questions

**Comments to the Author**

1. Is the manuscript technically sound, and do the data support the conclusions?

Reviewer #1: Yes

Reviewer #2: Yes

2. Has the statistical analysis been performed appropriately and rigorously? 

Reviewer #1: Yes

Reviewer #2: No

3. Have the authors made all data underlying the findings in their manuscript fully available?

Reviewer #1: Yes

Reviewer #2: Yes

4. Is the manuscript presented in an intelligible fashion and written in standard English?

Reviewer #1: Yes

Reviewer #2: Yes

5. Review Comments to the Author

Reviewer #1: 1. Data Availability & Reproducibility

• The paper states that data cannot be shared due to lack of informed consent. While this is understandable, the lack of even anonymized metadata or synthetic datasets limits reproducibility.

o Recommendation: Consider sharing preprocessed or synthetic data demonstrating the manipulation pipelines, or detailed code for FTO manipulation.

2. Participant Demographics and Bias

• All participants were employees of a hearing aid manufacturer (WS Audiology). This introduces potential bias due to professional exposure to hearing impairment.

o Recommendation: Include a discussion in the limitations section about how this might affect generalizability of perceptual ratings.

3. Ecological Validity of Conversations

• The use of short, task-based conversation snippets may not reflect real-life conversational dynamics.

o Recommendation: Acknowledge that findings may differ for emotionally rich or spontaneous conversations. Templeton & Wheatley (2023) is rightly cited—consider expanding this discussion.

4. Gender Interaction (Ease-of-Following Ratings)

• The observed gender effect lacks theoretical backing and is somewhat speculative.

o Recommendation: Either minimize emphasis on this finding or contextualize it with prior literature on gender differences in speech perception, if available.

5. Terminology Clarity

• The concept of "FTO" is central but could be better introduced.

o Recommendation: A simple diagram early in the manuscript (e.g., in Figure 1) with speaker turns and FTO visualized would greatly help readers unfamiliar with this terminology

Reviewer #2: In the present study, the author demonstrated that variations in turn-taking time are influenced by hearing loss. However, the manuscript requires significant revisions before it can be considered for publication in PLOS ONE.A major revision is recommended.

1.In the statistical analysis, the author does not consider the impact of medication and environmental factors. Please clarify why these variables were excluded and discuss their potential effects on the study outcomes.

2.In the Introduction section, the author is advised to clearly highlight the novelty of the present study in the context of existing literature.

3.The manuscript should address the influence of speech levels and signal-to-noise ratio on the results.

4.Avoid using bold and italic text in the manuscript.

5.Modify the reference style according to the journal guidelines.

6.Define abbreviations when they are first introduced in the manuscript.

6. PLOS authors have the option to publish the peer review history of their article (what does this mean? ). If published, this will include your full peer review and any attached files.

**Do you want your identity to be public for this peer review?** For information about this choice, including consent withdrawal, please see our Privacy Policy .

Reviewer #1: No

Reviewer #2: No

---

## [Author Response · Author response to Decision Letter 1]

1 May 2025

Dear Reviewers

Thank you for your feedback on the manuscript “Hearing-Lo ss Related Variations in Turn-Taking Time affect how Conversations are Perceived”. A reoccurring comment relates to making the data publicly available and I realize that the community have different opinions regarding what is allowed to publish without the consent of the participants: Our lab s of the opinion that when participants are not informed and have consented to having their (anonymized) data made available, we cannot readily share it. However, as we see that it is increasingly becoming a requirement to publish data, we have chosen to make the data of the current study available via Zenodo (DOI 10.5281/zenodo.15307719.). Furthermore, I have updated the Competing Interests statement in the submission of the review.

Below is a table of each comment and the I have addressed the individual comments and the alterations made to the resubmitted manuscript.

I hope with the improvements made You will consider the manuscript for publication in PLOS One.

Eline Borch Petersen

Editor Comments:

Please ensure that your manuscript meets PLOS ONE's style requirements, including those for file naming. The PLOS ONE style templates can be found at https://journals.plos.org/plosone/s/file?id=wjVg/PLOSOne_formatting_sample_main_body.pdf and https://journals.plos.org/plosone/s/file?id=ba62/PLOSOne_formatting_sample_title_authors_affiliations.pdf

REPLY

The numbering of headings has been removed, the citation style changed, and the author affiliation has been updated.

Thank you for stating the following in the Competing Interests section:

The author is an employee at the hearing-aid manufacturing company WS Audiology

REPLY

Thank you, the I will add the statement to the Competing Interests statement

REVIEWER 1

Data Availability & Reproducibility :The paper states that data cannot be shared due to lack of informed consent. While this is understandable, the lack of even anonymized metadata or synthetic datasets limits reproducibility.

Recommendation: Consider sharing preprocessed or synthetic data demonstrating the manipulation pipelines, or detailed code for FTO manipulation.

REPLY

The subjective rating data has been made available in anonymized form for via Zenodo (10.5281/zenodo.15307719.) together with an example script of how the FTOs are manipulated.

The following has been added: “An example script of the Matlab-implementation of the manipulation steps can be found on Zenodo (10.5281/zenodo.15307719).”

Participant Demographics and Bias: • All participants were employees of a hearing aid manufacturer (WS Audiology). This introduces potential bias due to professional exposure to hearing impairment.

Recommendation: Include a discussion in the limitations section about how this might affect generalizability of perceptual ratings.

REPLY

Thanks for the comment, which I would normally be happy to comply with, however for this study I do not find the bias relevant.

Firstly, although some employees are trained audiologists and/or have regular interaction with hearing impaired people, a vast majority of my colleagues in e.g. HR, finance and even R&D have never interacted professionally with a hearing-impaired person. Secondly, the third-party listeners were not informed of the experimental conditions tested and most of them are not aware of the underlying research I and others have conducted regarding the effect of hearing impairment on turn-taking timing.

I can see that a bias can occur if the participants were actively participating in conversations or if the talkers in the conversations they were listening to displayed signs of being older or mishearing/understanding, however none of this was the case in the current study. In our study all talkers were younger and communicating in their native language in quiet.

Hence, I cannot see to what extend being employed at WS Audiology would create a bias in the subjective ratings of the flow, naturalness and ease-of-following in the current study.

3. Ecological Validity of Conversations

• The use of short, task-based conversation snippets may not reflect real-life conversational dynamics.

o Recommendation: Acknowledge that findings may differ for emotionally rich or spontaneous conversations. Templeton & Wheatley (2023) is rightly cited—consider expanding this discussion.

REPLY

I have added details to the discussion on the differences between the conversation used by Templeton and the ones used in the current study. However, as the results are concurrent between the current study and Templeton, I see no reason to ´dive deeper into the difference in speech material, especially considering that Templeton manipulated only six short conversation segments in their study and that the attributes rated do not concur between our studies.

Text added to the discussion:

“When comparing between studies it should be noted that the conversations manipulated by Templeton consisted of six segments of 10 turns extracted from spontaneous conversations, whereas the 32 segments consisting of between 12-25 turns used in the current study were extracted from conversations driven by solving a task. Despite the difference in the nature of the conversations, the results from the current study, compliments the findings by Templeton that indeed, the FTO median times seem to affect the third-party perception of conversations more than the FTO variability.”

4. Gender Interaction (Ease-of-Following Ratings)

• The observed gender effect lacks theoretical backing and is somewhat speculative.

o Recommendation: Either minimize emphasis on this finding or contextualize it with prior literature on gender differences in speech perception, if available.

REPLY

I have tried to minimize the emphasis of the gender difference, exactly because it is not ‘easily’ explained. Although I have not been able to find theoretical backing for the effect, it is supported by statistical significance, which I personally think it weights higher than the presence of existing scientific evidence.

I have removed the following passage from the discussion:

“In the search of relevant scientific evidence, studies have focused on how gender affects active participation in conversations (being an interlocutor), but no studies have focused on how gender affects the evaluation and perception of conversations. Although a detailed description of the ease-of-following question was provided to the listeners, the gender difference suggests that there must be a gender-driven difference in how the ease-of-following is evaluated, to the extent that female listeners exhibit a ceiling effect in their ratings across all FTO conditions (Fig 2B)”

5. Terminology Clarity

• The concept of "FTO" is central but could be better introduced.

o Recommendation: A simple diagram early in the manuscript (e.g., in Figure 1) with speaker turns and FTO visualized would greatly help readers unfamiliar with this terminology

REPLY

Figure 1 has been altered to also include the FTO values of the original speech. Furthermore, the following elaboration has been added to the introduction:

“When contributing to a conversation, the verbal utterance is often delivered in response to what your conversation partner has just said. However, when you start your turn is not completely without importance, which is what we will explore in this study. The timing of a turn is denoted a floor-transfers offsets (FTOs), i.e. the time between the conversation partner stop talking and you start, in other words, the floor is transferred from one talker to another. This FTO is measured in milliseconds and will be negative if you and your partner are talking in an overlap and positive if there is a pause between the conversation partner stops and you start”

REVIEWER 2

1.In the statistical analysis, the author does not consider the impact of medication and environmental factors. Please clarify why these variables were excluded and discuss their potential effects on the study outcomes.

REPLY

I am afraid I do not understand this comment.

Is it referring to medication taken by the third-party listeners how are rating the conversations? I have never heard that medication should impact the subjective perception of speech. And if it does, I would assume it that the effect would not change throughout the experiment in which case adding subject as a random effect to the statistical model would account for the difference between participants.

Which environmental factors is referred to here? Is it the environment that people are in when they perform the online experiment?

Nevertheless, I have no information on either, so I unable to account for them in the statistical analysis.

2.In the Introduction section, the author is advised to clearly highlight the novelty of the present study in the context of existing literature.

REPLY

A sentence has been added to the introduction to further highlight the difference and novelty of this study:

“The purpose of the current study is to investigate who naturally occurring variations in both FTO median, as well as the often overlooked, FTO IQR affects how conversations are perceived”

3.The manuscript should address the influence of speech levels and signal-to-noise ratio on the results.

REPLY:

The rms of each talker within a conversation segment was equalized to ensure that the speech levels were the same across talkers and conversation segments. The participants were instructed and given an example of a conversation before the actual test, and asked to adjust the volume to a comfortable level, which I assume result in the speech being highly intelligible. I have no information regarding the absolute presentation level, as is the case with most online hearing experiments.

4.Avoid using bold and italic text in the manuscript.

REPLY

I have removed the bold writing when referring to figures and the tables, the remainder is kept to the improve readability of the text.

5.Modify the reference style according to the journal guidelines.

REPLY

Thank you, the citation style and references to the figure are now changed

6.Define abbreviations when they are first introduced in the manuscript.

REPLY

I have gone through the manuscript again and cannot find any case where this is not done.

---

## [Decision Letter · Decision Letter 1]

12 May 2025

Hearing-Loss Related Variations in Turn-Taking Time affect how Conversations are Perceived

PONE-D-25-12104R1

Dear Dr. Petersen,

We’re pleased to inform you that your manuscript has been judged scientifically suitable for publication and will be formally accepted for publication once it meets all outstanding technical requirements.

Kind regards,

Niravkumar Joshi

Academic Editor

PLOS ONE

Additional Editor Comments (optional):

Reviewers' comments:

Reviewer's Responses to Questions

**Comments to the Author**

1. If the authors have adequately addressed your comments raised in a previous round of review and you feel that this manuscript is now acceptable for publication, you may indicate that here to bypass the “Comments to the Author” section, enter your conflict of interest statement in the “Confidential to Editor” section, and submit your "Accept" recommendation.

Reviewer #1: All comments have been addressed

Reviewer #2: (No Response)

2. Is the manuscript technically sound, and do the data support the conclusions?

Reviewer #1: Yes

Reviewer #2: Partly

3. Has the statistical analysis been performed appropriately and rigorously? 

Reviewer #1: Yes

Reviewer #2: Yes

4. Have the authors made all data underlying the findings in their manuscript fully available?

Reviewer #1: Yes

Reviewer #2: Yes

5. Is the manuscript presented in an intelligible fashion and written in standard English?

Reviewer #1: Yes

Reviewer #2: Yes

6. Review Comments to the Author

Reviewer #1: This revised manuscript effectively addresses prior reviewer comments and adds important clarification, particularly through the addition of Figure 1 and clearer articulation of Floor Transfer Offset (FTO) concepts. The methodological approach is strong and reproducible, and the use of publicly available data and code strengthens transparency.

Minor suggestions:

Please ensure all figures are provided at the highest possible resolution for publication.

Consider rewording a few phrases in the Results section for smoother flow.

A more explicit statement on limitations (e.g., sample population from hearing aid company) would benefit the Discussion.

Overall, this is a clear, well-structured, and methodologically sound article that adds value to the field of hearing-related communication studies

Reviewer #2: (No Response)

7. PLOS authors have the option to publish the peer review history of their article (what does this mean? ). If published, this will include your full peer review and any attached files.

**Do you want your identity to be public for this peer review?** For information about this choice, including consent withdrawal, please see our Privacy Policy .

Reviewer #1: No

Reviewer #2: No

---

## [Editor Report · Acceptance letter]

PONE-D-25-12104R1

PLOS ONE

Dear Dr. Petersen,

I'm pleased to inform you that your manuscript has been deemed suitable for publication in PLOS ONE. Congratulations! Your manuscript is now being handed over to our production team.

Kind regards,

on behalf of

Dr. Niravkumar Joshi

Academic Editor

PLOS ONE